# Lessons from Media-Centered Climate Change Literature

**Cjyar Nazar Dhaher Dhaher \* and Agah Gümüş**

Department of Communication and Media Studies, Eastern Mediterranean University, Famagusta 99628, Turkey; agah.gumus@emu.edu.tr
\* Correspondence: cjyar_nazar@yahoo.com

**Abstract:** Climate change has become a global challenge over the years and the media has played an important part in the dissemination of information, research on which constitutes a substantial body of scientific literature. This study aimed to map media-centered climate change articles to highlight the relevant lessons within the field. The lessons learned are as follows. (1) Two environmental communication journals are the primary publication venues that prioritize climate change-focused articles. (2) Media-centered climate change literature peaked in 2020, and (3) many countries attracted scholarly interest. (4) Newspapers were examined more than any other media form. (5) The first authors of media-centered climate change articles are westerners. (6) Almost 50% of media-centered climate change articles did not adopt a theory. (7) Quantitative and (8) content analysis methods are the most adopted data collection methods. (9) Finally, this review shows that communication and media scholars are most interested in media-centered climate change articles.

**Keywords:** climate change; media; systematic literature review; literature

## 1. Aim and Background

Over the last few decades, the field of climate change has been a subject of intense discourse within various levels of society, since climate change is not merely an environmental risk, but the greatest threat to human existence. To highlight its severity, the United Nations called it "the defining issue of our time"; this is because our world has been experiencing unprecedented and catastrophic events such as intense heatwaves, droughts, storms, rising sea levels, and melting glaciers, all as a result of climate change (https://www.un.org/en/sections/issues-depth/climate-change/?utm_medium=website&utm_source=archdaily.com.br (accessed on 28 May 2019).

For the layman, the discourse of climate change is inconceivable in the first instance, because climate change discourses consist of discussions around carbon dioxide ($CO_2$), greenhouse gases, fossil fuels, emissions, sea-level rise, global average temperature, and renewable energy. Many of the above aspects have affected the language of everyday people. Secondly, the discourse takes place at an intergovernmental level, so it is quite removed from everyday people and their lives. As a result, a significant number of people learn about climate change issues and events from various media sources.

The media continues to play a critical role in coverage as well as framing of the issues, people, and events in and around climate change. According to Wozniak, Lück, and Wessler [1] the mass media has a mission to report novelties, reduce complexity, and tell engaging stories. When an issue is long-term, multi-faceted, and largely unobtrusive—as is the case with climate change—these ambitions create challenges for news production. One journalistic mechanism to respond to such challenges is the use of narrative or visual "hooks" on which to "hang" a news report. The resulting combination of elements—facts and statements, story structure, and images—can lead to journalistic articles in which the configuration of these elements seems incoherent at first [1].

In the academic context, the use of the two concepts "media" and "climate change", which form the basis of this field, is not new to academic articles, books, book chapters,

or other academic resource forms. As acknowledged by many scholars, it can be said that a substantial body of scientific literature exists on media-centered climate change, ranging from the media framing of climate change in Peru [2] and Canadian, American and international [3] miscommunication regarding climate change [4,5] to the ownership structure of media organizations and its impact on climate reporting [6]. Internet memes related to climate change [7] have received scholars' attention as well, both within media studies and environmental communication. Although media and climate change cannot yet be considered a distinct discipline, it is important to describe the state of the field first, and, in parallel, to underline the obvious long-standing relationship between the media and climate action, since scholars have focused on even smaller aspects of climate change literature, such as Björnberg et.al [8], who explored the peer-reviewed literature on the denial of climate change through a meta-analysis of climate change denial articles published on Web of Science and Scopus between 1990 and 2015. Finally, the topic is important because of the ongoing coverage of climate change across various media platforms around the world and the continuous publication of media-related issues within climate change.

Taking into account the continuous media-related activity within climate change literature, it is necessary to provide an overview of the publication avenues of media-related climate change-centered articles, because such information provides us with the journals that account for the highest percentage of published media-related climate change-centered articles. Climate change is a complex topic that involves complicated science and knowledge of a wide range of scientific domains [9] with many variables interrelating over time [10] making precise predictions problematic [11]. Given the purpose of this study, information was seen as a necessity in research within the sub discipline, because providing information on the research area will show where there is increased academic interest or a lack thereof over the period under consideration within this field. As reported by Baumeister and Leary [12], a high-quality systematic literature review should provide opportunities for theory development and describe reoccurring research topics within the field. Therefore, this study aimed to provide information on the country of the first or corresponding authors' university, and to provide an overview of the dominant methodological approaches adopted within this field. This will enable the reader to see the progression of media analyzed within the topic of media-related climate change. Overall, the primary aim of this study is to descriptively assess the bibliographical information of media-related climate change-centered literature by using peer-reviewed articles within Web of Science, the foremost index for the natural and social sciences. This review is organized around ten research statements:

- Publication venue;
- Dominant subdiscipline of publication venue;
- Theory;
- Reoccurring topics;
- Host country of the university of the first or corresponding author;
- First or corresponding author's department;
- Methodological approaches;
- Analytical approaches;
- Media analyzed;
- Year of publication.

## 2. Method

### 2.1. Systematic Review

To perform a media-related climate change bibliographic overview, this study adopted a systematic review (sometimes called a meta-analysis or a compendium study), which is the method of inquiry that is (1) considered the clearest and the most concise means of creating a big-picture view of the state of research, (2) one of the most useful means of tracing the features of peer-reviewed literature, (3) the best means of analyzing the methods and data collection techniques adopted in the literature, (4) able to reveal the

trends as well as oversights in the literature, and (5) used to map the progression of similar fields and reveal interesting findings. For example, scholars such as Comfort and Park [13] adopted a meta-analysis to understand the landscape of the peer-reviewed literature on environmental communication.

In line with information available online and to the best of our knowledge, as of 2021, the only available systematic literature review in this field is a study conducted by Schafer and Schlichting [14]. Although a very well-conducted study for its time, it is evident that a great deal has happened in the seven years since the study was conducted. The study was also largely limited in its scope. A total of 133 peer-reviewed articles was reviewed. The study focused exclusively on the progression of the field, the analytical approaches adopted, countries, types of media, and the methodological approaches adopted within the field. The current study was conceived to address these limitations; however, we seek to not only improve these findings, but to also address other important questions, such as the theories are adopted within the field, the first or corresponding author's department, the country of the first or corresponding author's university, publication venues, and the dominant sub discipline of the publication venues.

Furthermore, although there is no generally accepted method of conducting a systematic literature review, we adopted Denyer and Tranfield's [15] method, which consists of the following steps: (1) research question formulation, (2) locating academic resources, (3) assessing qualified articles, (4) analysis and synthesis, and (5) reporting findings. Our study discusses the research question formulation in the introductory section; in addition, the location of the academic resources and assessment of the qualified articles are discussed in this section. We carry out the analysis and reporting of findings in the Analysis section.

## 2.2. Selection of Databases, Coding, and Bibliographic Limitations

Given that the two important keywords, "media" and "climate change", that form the basis of this study are multidisciplinary, this study adopted the Web of Science database to collect media-related climate-change-focused articles. We chose Web of Science (http://webofknowledge.com; https://clarivate.com/webofsciencegroup/solutions/web-of-science/ (accessed on 8 March 2019); "The Web of Science™ is the world's most trusted publisher-independent global citation database. Guided by the legacy of Dr Eugene Garfield, inventor of the world's first citation index, the Web of Science is the most powerful research engine, delivering your library with best-in-class publication and citation data for confident discovery, access and assessment. Our multidisciplinary platform connects regional, specialty, data and patent indexes to the Web of Science Core Collection™. Our comprehensive platform allows you to track ideas across disciplines and time from almost 1.9 billion cited references from over 171 million records [13]. Since it is the largest collection of reputable journals in the social sciences and natural sciences as mentioned by Comfort and Park [13], the database contains over 18,000 highly respected academic resources published from 1975. Many are relevant to our focus, including the *Journal of Communication*, *Communication Research*, and *Mass Communication and Society*, as well as journals likely to contain articles specific to issues of environmental communication, such as *Environmental Communication*, *Science Communication*, and *Public Understanding of Science* [13].

We coded the first authors' university country by following Schafer and Schlichting's [14] method for analyzing studies within climate change; hence, the following regions were found: the UK, Germany, France, Sweden Russia, Canada, Mexico, India, Middle East, China, Japan, Oceania, Australia, New Zealand, Brazil, Argentina, Africa, South Africa. Regarding methodological approaches, we coded them as qualitative methods, quantitative methods, and mixed methods. Regarding analytical approaches, we adopted Comfort and Park's protocol [13] and considered rhetorical analysis, content analysis, surveys, narrative essays, experiments, historical analysis, interviews, case studies and focus groups, ethnographies, and secondary data analysis. For media, we adopted Comfort and Park [13] and Schafer and Schlichting's [14] methods and coded the following outlets: newspapers, magazines, television, radio, TV news, TV serials, films, documentaries, internet, websites,

social media, and search engines, websites of NGOs, books, and advertisements. Regarding the year of publication, we coded from 1975 to 2020. For the sub discipline of publication venues, we coded communication media studies, environmental sciences, general, inter-disciplinary, and others. Finally, for every media-related climate change paper, the theory adopted, reoccurring topics, and publication venues were analyzed.

In order to obtain a media-related climate change bibliographic overview, we narrowed our search to two keywords: "media" and "climate change". To ensure the qualification of the articles, we considered whether they mentioned "media" and "climate change" in their title, abstract, and keywords. Our preliminary search produced 303 media-related climate change articles; after limiting our document type to journal articles, 232 articles remained. After perusing the articles, we found that 49 of the articles only mentioned media in two of the three parts of the study, therefore leaving us with 183 articles; six articles were focused on geological media—for example, Bachu and Adams [16] "Sequestration of $CO_2$ in geological media in response to climate change: road map for site selection using the transform of the geological space into the $CO_2$ phase space"—which are entirely different from media studies-focused climate change papers. Out of 169 articles, 10 of the articles were not available online. As a result, the final number of media-centered studies for this current study was 167 (Figure 1).

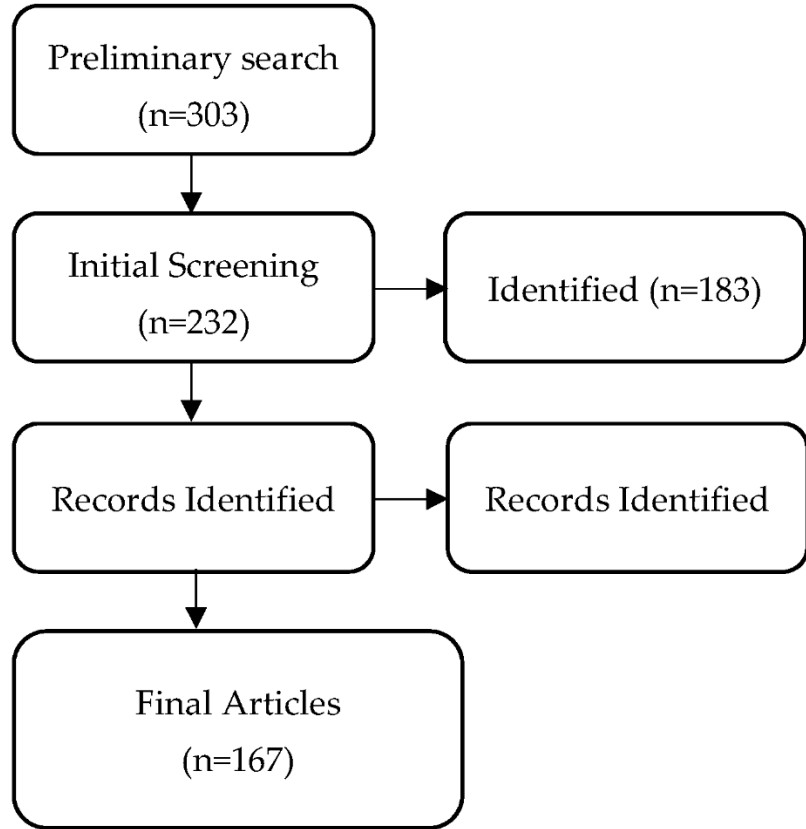

**Figure 1.** Flow chart of resource selection.

### 2.3. Data Collection, Data Entry Process

For each of the qualified media-related climate change-focused articles obtained for this study, we exported the title, first or corresponding author's affiliations, publication venues, year of publication, and affiliations (journal name), and we coded the identified records for the dominant sub discipline of the publication venue, theory adopted, reoccurring topics, first author's university's host country, methodological approaches, analytical approaches, and media analyzed, all of which were exported to an MS Excel spreadsheet. Data were inputted by two researchers in the field of media and public relations who

screened the articles. During the data entry phase, disagreements were resolved through consensus. After the completion of the data collection in Excel, data were input into SPSS for the generation of frequencies and percentage distributions.

*2.4. Limitation*

This study had some limitations, many of which were methodological. To define our search, we focused on two major keywords and, as a result, media-related climate change papers that did not use these keywords may not have been included in our sample. In addition, it is possible that a few media-related climate change papers were published before 1975; however, as highlighted above, the Web of Science archives are also limited in terms of the year of publication. As Comfort and Park [13] stated, "while the Web of Science is frequently used as a source for studies of this type, it has its own limitations, including an emphasis on English-language scholarship".

**3. Results**

In this section, we present the results, focusing on publication venues, the dominant sub discipline of publication venues, theory adopted, reoccurring topics, first author's university's host country, methodological approaches, analytical approaches, media analyzed, and year of publication, respectively. We also discuss the results by placing our findings within the context of the available literature.

*3.1. Lesson 1: Two Environmental Communication Journals Are the Primary Publication Venues Prioritizing Climate Change-Focused Articles*

*Environmental Communication: A Journal of Nature and Culture* and *Public Understanding of Science* are the two main publications for media-related climate-change-focused articles (n = 39). Other journals that published more than two articles are *Environmental Research Letters, Journal of Public Economics, Energy Research & Social Science, Current Issues in Tourism, Media Education—Mediaobrazovanie, Revista Latina de Comunicacion Social, International Communication Gazette, Convergencia Revista de Ciencias Sociales, Transactions of the Institute of British Geographers, Canadian Journal of Sociology—Cahiers Canadiens de Sociologie, Society & Natural Resources, Canadian Review of Sociology—Revue Canadienne de Sociologie,* and *British Journal of Politics & International Relations* (*n* = 28). Apart from *Revista Mediterranea Comunicacion—Journal of Communication*, other journals that only published a singular media-related climate change article were the *International Journal of Communication, Kotuitui—New Zealand Journal of Social Sciences Online, Progress in Nuclear Energy, Journalism, Environmental Science and Pollution Research, Alternatives, Frontiers in Ecology and Evolution, Journalism Practice, International Journal of E-Politics, African Journalism Studies, Communication Research, Global Environmental Politics, Environment Development and Sustainability, Convergence—The International Journal of Research Into New Media Technologies, Environmental Values, Communications—European Journal of Communication Research, Mass Communication and Society, Security Dialogue, Cogent Arts & Humanities, Space and Polity, Jamba—Journal of Disaster Risk Studies, Meridiano Journal of Global Studies, Energy & Environment, Cuadernos Info, Human and Ecological Risk Assessment, Journal of European Integration, Environmental Science & Policy, triple C: Communication, Capitalism & Critique, American Behavioral Scientist, South African Journal on Human Rights, Filosofija. Sociologija, Journal of Media Psychology—Theories Methods and Applications, Ps—Political Science & Politics, Cultural Geographies, Political Research Quarterly, Bioscience, Human Ecology Review, Wiley Interdisciplinary Reviews—Climate Change, International Journal of Climate Change Strategies and Management, Risk Analysis, Interdisciplinary Science Reviews, International Journal of Science Education,* and *Revista Mediterranea Comunicacion—Journal of Communication*.

As for subdisciplines of Table 1, we found that the majority of the articles were published within media studies publications (n = 49). Environmental sciences and interdisciplinary articles were published in (n = 45) and (n = 18), respectively.

**Table 1.** Publication venues and number of articles.

| Journals | No of Articles |
| :---: | :---: |
| *Envi. Comm.: A Journal of Nature and Culture* | 25 |
| *Public Understanding of Science* | 14 |
| *Global Envi. Change—Human & Policy Dimensions* | 8 |
| *Climatic Change* | 7 |
| *Science Communication* | 6 |
| *Journalism Studies* | 5 |
| *Pacific Journalism Review* | 4 |
| *Energy Policy* | 3 |
| *Environmental Politics* | 3 |
| *Environmental Research Letters* | 2 |
| *Journal of African Media Studies* | 2 |
| *Chasqui. Revista Latinoamericana de Comunicacion* | 2 |
| *Journal of Public Economics* | 2 |
| *Energy Research & Social Science* | 2 |
| *British Journal of Politics & International Relations* | 2 |
| *Desenvolvimento e Meio Ambiente* | 2 |
| *Transactions of the Institute of British Geographers* | 2 |
| *Canadian Review of Sociology—Revue Canadienne de Sociologie* | 2 |
| *European Journal of Communication* | 2 |
| *Society & Natural Resources* | 2 |
| *Canadian Journal of Sociology—Cahiers Canadiens de Sociologie* | 2 |
| *Filosofija. Sociologija* | 2 |
| *Current Issues in Tourism* | 2 |
| *Media Education—Mediaobrazovanie* | 2 |
| *Revista Latina de Comunicacion Social* | 2 |
| *Redes.com: Revista de estudios para el desarrollo social de lacomunicacion* | 2 |
| *International Communication Gazette* | 2 |
| *Revista Mediterranea Comunicacion—Journal of Communication* | 1 |

*3.2. Lesson 2: Media-Centered Climate Change Literature Peaked in 2020*

As for the progression of the media-centered climate change literature, the result shows an upward trend for the eighteen years covered by our sample. Table 2 shows that 2020 and 2019 had the highest number of media-centered climate change articles (n = 30%). Moreover, apart from 2018, 2017, 2016, 2015, and 2014, all other years had less than ten media-centered climate change articles.

**Table 2.** The number of media-centered climate change articles from 1998 to 2020.

| | Frequency | Percent | Valid Percent | Cumulative Percent |
| :---: | :---: | :---: | :---: | :---: |
| 2020 | 26 | 15.0 | 15.0 | 15.6 |
| 2019 | 25 | 15.0 | 15.0 | 30.5 |
| 2018 | 17 | 10.2 | 10.2 | 40.7 |
| 2017 | 17 | 10.2 | 10.2 | 50.9 |
| 2016 | 14 | 8.4 | 8.4 | 59.3 |
| 2015 | 16 | 9.6 | 9.6 | 68.9 |
| 2014 | 13 | 7.8 | 7.8 | 76.6 |
| 2013 | 9 | 5.4 | 5.4 | 82.0 |
| 2012 | 8 | 4.8 | 4.8 | 86.8 |
| 2011 | 4 | 2.4 | 2.4 | 89.2 |
| 2010 | 8 | 4.8 | 4.8 | 94.0 |
| 2009 | 5 | 3.0 | 3.0 | 97.0 |

**Table 2.** *Cont.*

|  | **Frequency** | **Percent** | **Valid Percent** | **Cumulative Percent** |
|---|---|---|---|---|
| 2008 | 1 | 0.6 | 0.6 | 97.6 |
| 2007 | 1 | 0.6 | 0.6 | 98.2 |
| 2005 | 1 | 0.6 | 0.6 | 98.8 |
| 2000 | 1 | 0.6 | 0.6 | 99.4 |
| 1998 | 1 | 0.6 | 0.6 | 100.0 |
| Total | 167 | 100.0 | 100.0 | |

With up to four media-centered climate change studies published per year between 2009 and 2020, the number of articles remained moderate until 2013. From 2014, media-focused climate change publications increased significantly. A total of 25 articles were published in 2019, while 17 were published in 2018 and 2017. As for 2016, 2015, and 2014, a total of 14, 16, and 13 articles were published, respectively.

*3.3. Lesson 3: Media-Centered Climate Change Articles Focused on Multiple Countries as Well as the United States and United Kingdom*

Table 3 results show that the majority of the media-related climate change articles focused on more than two countries (n = 22.2%). Furthermore, the countries that were most frequently examined within the media-centered climate change articles were the United States and the United Kingdom (22.2%). In addition, 6% of the articles did not focus on any country, while Canada, India, Australia, Germany, Spain, Sweden, Ireland, Singapore, Malaysia, Tanzania, the Netherlands, Italy, Poland, Russia, and Brazil were the focus of more than one article (n = 71). Other countries that were the focus of only one media-related climate change article were Japan, China, Uganda, Pakistan, Croatian, South Korea, Zimbabwe, France, South Africa, Chile, Argentina, Peru, Switzerland, Belgium, the Czech Republic, Turkey, Ghana, Denmark, Greenland, Chile, and Vietnam (n = 21; 12.6)

**Table 3.** Countries represented in studies.

|  | **Frequency** | **Percent** | **Valid Percent** | **Cumulative Percent** |
|---|---|---|---|---|
| United States | 23 | 13.8 | 13.8 | 13.8 |
| United Kingdom | 14 | 8.4 | 8.4 | 22.2 |
| China | 1 | 0.6 | 0.6 | 22.8 |
| Australia | 6 | 3.6 | 3.6 | 26.3 |
| Canada | 8 | 4.8 | 4.8 | 31.1 |
| Spain | 5 | 3.0 | 3.0 | 34.1 |
| Sweden | 5 | 3.0 | 3.0 | 37.1 |
| Germany | 6 | 3.6 | 3.6 | 40.7 |
| India | 7 | 4.2 | 4.2 | 44.9 |
| Brazil | 3 | 1.8 | 1.8 | 46.7 |
| Japan | 1 | 0.6 | 0.6 | 47.3 |
| No country | 10 | 6.0 | 6.0 | 53.3 |
| Multiple Countries | 38 | 22.8 | 22.8 | 76.0 |
| Italy | 2 | 1.2 | 1.2 | 77.2 |
| Uganda | 1 | 0.6 | 0.6 | 77.8 |
| Ireland | 4 | 2.4 | 2.4 | 80.2 |
| Poland | 3 | 1.8 | 1.8 | 82.0 |

**Table 3.** *Cont.*

|  | Frequency | Percent | Valid Percent | Cumulative Percent |
|---|---|---|---|---|
| Pakistan | 1 | 0.6 | 0.6 | 82.6 |
| Croatian | 1 | 0.6 | 0.6 | 83.2 |
| Russia | 2 | 1.2 | 1.2 | 84.4 |
| Netherlands | 2 | 1.2 | 1.2 | 85.6 |
| Singapore | 4 | 2.4 | 2.4 | 88.0 |
| Tanzania | 2 | 1.2 | 1.2 | 89.2 |
| Switzerland | 1 | 0.6 | 0.6 | 89.8 |
| Belgium | 1 | 0.6 | 0.6 | 90.4 |
| Czech Republic | 1 | 0.6 | 0.6 | 91.0 |
| Turkey | 1 | 0.6 | 0.6 | 91.6 |
| Ghana | 1 | 0.6 | 0.6 | 92.2 |
| Denmark | 1 | 0.6 | 0.6 | 92.8 |
| Greenland | 1 | 0.6 | 0.6 | 93.4 |
| Chile | 1 | 0.6 | 0.6 | 94.0 |
| Vietnam | 1 | 0.6 | 0.6 | 94.6 |
| Malaysia | 2 | 1.2 | 1.2 | 95.8 |
| South Korea | 1 | 0.6 | 0.6 | 96.4 |
| Zimbabwe | 1 | 0.6 | 0.6 | 97.0 |
| France | 1 | 0.6 | 0.6 | 97.6 |
| South Africa | 1 | 0.6 | 0.6 | 98.2 |
| Chile | 1 | 0.6 | 0.6 | 98.8 |
| Argentina | 1 | 0.6 | 0.6 | 99.4 |
| Peru | 1 | 0.6 | 0.6 | 100.0 |
| Total | 167 | 100.0 | 100.0 |  |

*3.4. Lesson 4: Media-Centered Climate Change Articles Examined Newspapers More Than Any Other Media Form*

Based on our quantitative data in Table 4, the findings show that newspapers were distributed more widely than the combination of all other media outlets explored within the media-centered climate change literature (68.9%). Social networking sites and the internet followed, with 31 (18.6%) articles addressing them, while TV news, websites and search engines accounted for 17 media-centered climate change articles. Magazines, documentaries, and advertising represented the least prevalent media forms, at 1.2%, 0.6% and 0.6%, respectively.

*3.5. Lesson 5: First or Corresponding Authors of Media-Centered Climate Change Articles Are Mostly Hosted in Western and Developed Nations*

The quantitative results of Table 5 showed that the majority of the media-related climate change articles were published by students, researchers and/or academics in two countries: the United States of America and the United Kingdom (n = 48). Following these two, the countries with students, researchers and/or academics that wrote the most media-centered climate change articles were Canada, Spain, Australia, Sweden, and Germany (n = 53). Only one media-related climate change article was published by students, researchers and/or academics in the following countries: China, India, Pakistan, Croatia, Russia, Austria, Czech Republic, Turkey, Ghana, Chile, Norway, Chile, Zimbabwe, and Mexico (n = 14). It was also observed that 89.8 percent of the first authors of the studies

were also the corresponding authors of the studies (150 out of 167). Therefore, in the calculations, we refer to the first or corresponding authors of the study together in table legends, and we did not take into consideration the 17 papers in which the authors were neither the first nor the corresponding authors.

**Table 4.** Media represented in studies.

| | Frequency | Percent | Valid Percent | Cumulative Percent |
|---|---|---|---|---|
| Newspapers | 115 | 68.9 | 68.9 | 68.9 |
| Magazines | 2 | 1.2 | 1.2 | 70.1 |
| TV News | 7 | 4.2 | 4.2 | 74.3 |
| Documentary | 1 | 0.6 | 0.6 | 74.9 |
| Internet | 10 | 6.0 | 6.0 | 80.8 |
| Websites | 6 | 3.6 | 3.6 | 84.4 |
| Social Media | 21 | 12.6 | 12.6 | 97.0 |
| Search Engines | 4 | 2.4 | 2.4 | 99.4 |
| Advertisements | 1 | 0.6 | 0.6 | 100.0 |
| Total | 167 | 100.0 | 100.0 | |

**Table 5.** Host country of the first or corresponding author's university.

| | Frequency | Percent |
|---|---|---|
| United States | 24 | 16 |
| United Kingdom | 24 | 16 |
| China | 1 | 0.7 |
| Australia | 11 | 7.3 |
| Canada | 15 | 10 |
| Spain | 12 | 8 |
| Sweden | 8 | 5.3 |
| Germany | 7 | 4.7 |
| India | 1 | 0.7 |
| Japan | 2 | 1.3 |
| Italy | 2 | 1.3 |
| Uganda | 3 | 2 |
| Portugal | 2 | 1.3 |
| Ireland | 2 | 1.3 |
| Poland | 4 | 2.7 |
| Finland | 2 | 1.3 |
| Pakistan | 1 | 0.7 |
| Croatian | 1 | 0.7 |
| Russia | 1 | 0.7 |
| Austria | 1 | 0.7 |
| Netherlands | 3 | 2 |
| Singapore | 2 | 1.3 |
| Tanzania | 2 | 1.3 |
| Switzerland | 4 | 2.7 |

**Table 5.** *Cont.*

|  | Frequency | Percent |
|---|---|---|
| Belgium | 3 | 2 |
| Czech Republic | 1 | 0.7 |
| Turkey | 1 | 0.7 |
| Ghana | 1 | 0.7 |
| Denmark | 3 | 2 |
| Zimbabwe | 1 | 0.7 |
| Mexico | 1 | 0.7 |
| France | 2 | 1.3 |
| Norway | 1 | 0.7 |
| Chile | 1 | 0.7 |
| Total | 150 | 100.0 |

*3.6. Lesson 6: Over 50% of Media-Centered Climate Change Articles Did Not Adopt a Theory*

The quantitative results of Table 6 showed that the majority of the media-centered climate change articles did not adopt a theory. Among studies that adopted theories (n = 79; 47.3%), Framing Theory dominated, with 15.6% of articles, while 12.6% adopted multiple theories. Only four other theories apart from the aforementioned were adopted more than once, including Cultural Theory, Discourse Theory, Informatics Theory, Justification Theory, Self-Efficacy Theory, and Uses and Gratification Theory (n = 15; 9%). Other theories that were adopted only once were as follows: Democratic Theory, Congruity Theory, Motivated Reasoning, Cognitive Theory, Social Identity Theory, Agenda Setting, Theory of Planned Behavior, Critical Race Theory, Narrative Genre Theory, MIT Theory, Critical Discourse Theory (CDA), Social Representations Theory (SRT), Gate Keeping Theory, Deliberative Theory, Conflict Theory, Field Theory, and Ecological Communication Theory (n = 10.2%).

**Table 6.** Theories represented in studies.

|  | Frequency | Percent | Valid Percent | Cumulative Percent |
|---|---|---|---|---|
| No Theory | 88 | 52.7 | 52.7 | 52.7 |
| Multiple Theories | 21 | 12.6 | 12.6 | 65.3 |
| Self-Efficacy Theory | 2 | 1.2 | 1.2 | 66.5 |
| Uses and Gratification | 2 | 1.2 | 1.2 | 67.7 |
| Framing Theory | 26 | 15.6 | 15.6 | 83.2 |
| Democratic Theory | 1 | 0.6 | 0.6 | 83.8 |
| Justification Theory | 2 | 1.2 | 1.2 | 85.0 |
| Congruity Theory | 1 | 0.6 | 0.6 | 85.6 |
| Motivated Reasoning | 1 | 0.6 | 0.6 | 86.2 |
| Cognitive Theory | 1 | 0.6 | 0.6 | 86.8 |
| Informatics Theories | 2 | 1.2 | 1.2 | 88.0 |
| Social Identity Theory | 1 | 0.6 | 0.6 | 88.6 |
| Agenda Setting | 1 | 0.6 | 0.6 | 89.2 |
| Discourse Theory | 3 | 1.8 | 1.8 | 91.0 |
| Theory of Planned Behavior | 1 | 0.6 | 0.6 | 91.6 |

**Table 6.** *Cont.*

|  | Frequency | Percent | Valid Percent | Cumulative Percent |
|---|---|---|---|---|
| Critical Race Theory | 1 | 0.6 | 0.6 | 92.2 |
| Narrative Genre Theory | 1 | 0.6 | 0.6 | 92.8 |
| MIT Theory | 1 | 0.6 | 0.6 | 93.4 |
| Critical Discourse Theory (CDA) | 1 | 0.6 | 0.6 | 94.0 |
| Social Representations Theory (SRT) | 1 | 0.6 | 0.6 | 94.6 |
| Cultural Theory | 4 | 2.4 | 2.4 | 97.0 |
| Gate Keeping Theory | 1 | 0.6 | 0.6 | 97.6 |
| Deliberative Theory | 1 | 0.6 | 0.6 | 98.2 |
| Conflict Theory | 1 | 0.6 | 0.6 | 98.8 |
| Field Theory | 1 | 0.6 | 0.6 | 99.4 |
| Ecological Communication Theory | 1 | 0.6 | 0.6 | 100.0 |
| Total | 167 | 100.0 | 100.0 | |

### 3.7. Lesson 7: Quantitative Methods Are the Most Adopted Methods within Media-Centered Climate Change Literature

The descriptive findings in Table 7 showed that the methods adopted within the media-centered climate change literature were almost equally distributed between qualitative and quantitative research methods; however, the majority of the media-centered climate change papers published between 1998 and 2020 adopted quantitative methods (48.5%). As for mixed methods, the results showed that only 10.8% of media-centered climate change articles adopted mixed methods.

**Table 7.** Methods represented in studies.

|  | Frequency | Percent | Valid Percent | Cumulative Percent |
|---|---|---|---|---|
| Qualitative | 68 | 40.7 | 40.7 | 40.7 |
| Quantitative | 81 | 48.5 | 48.5 | 89.2 |
| Mixed Methods | 18 | 10.8 | 10.8 | 100.0 |
| Total | 167 | 100.0 | 100.0 | |

### 3.8. Lesson 8: Content Analysis Is the Dominant Data Collection Method within Media-Centered Climate Change Literature

The findings of Table 8 showed that almost half of the media-centered climate change articles published between 1998 and 2020 adopted content analysis 45.5%. Rhetorical analysis and surveys followed 33.6%. Narrative essays, interviews, and case studies accounted for 1.8%, 4.2%, and 1.8%, respectively. As for data collection tools that were only adopted once in the media-centered climate change literature, these were found to be focus groups, ethnography, and secondary data analysis 1.8%.

### 3.9. Lesson 9: Although Almost Equally Distributed, Communication and Media Scholars Are More Interested in Media-Centered Climate Change Articles

The descriptive summary of Table 9 showed that, regarding the first author's department, scholars within the department of Communication and Media Studies published 50.7% of the media-centered climate change articles. Researchers, students and/or academics in the department of Environmental Sciences published only 33.3% of the media-centered climate change articles. Scholars in other fields of study also showed interest. Finally, interdisciplinary articles covering Communication and Media Studies and Environmental Sciences accounted for only 1.3% of the media-centered climate change articles.

**Table 8.** Data collection tools represented in studies.

|  | Frequency | Percent | Valid Percent | Cumulative Percent |
|---|---|---|---|---|
| Rhetorical Analysis | 34 | 20.4 | 20.4 | 20.4 |
| Content Analysis | 76 | 45.5 | 45.5 | 65.9 |
| Surveys | 22 | 13.2 | 13.2 | 79.0 |
| Narrative Essay | 3 | 1.8 | 1.8 | 80.8 |
| Experiments | 19 | 11.4 | 11.4 | 92.2 |
| Interviews | 7 | 4.2 | 4.2 | 96.4 |
| Case Studies | 3 | 1.8 | 1.8 | 98.2 |
| Focus Groups | 1 | 0.6 | 0.6 | 98.8 |
| Ethnography | 1 | 0.6 | 0.6 | 99.4 |
| Secondary Data Analysis | 1 | 0.6 | 0.6 | 100.0 |
| Total | 167 | 100.0 | 100.0 | |

**Table 9.** First author or corresponding author's department represented in the study.

|  | Frequency | Percent |
|---|---|---|
| Communication and Media Studies | 76 | 50.7 |
| Environmental Sciences | 50 | 33.3 |
| Interdisciplinary | 2 | 1.3 |
| Others | 22 | 14.7 |
| Total | 150 | 100.0 |

## 4. Discussing Lessons Learned

Taking into account the fact that communication and media are essential parts of many aspects of human life and are integral in different sectors of various human economies around the world, natural scientists have long acknowledged the role of the media in the coverage of climate-related issues and events. As reported by Schafer and Schlichting [14], "since the early 1990s, many studies have appeared which analyze how media present climate change to various audiences. The number of these studies has risen to a point at which a systematic review of the research field is warranted". Our current study generated 167 media-related climate change articles published between 1998 and 2020 and interesting results were found: some improving the first meta-analysis of this field by Schafer and Schlichting [14], as promised, and others supporting it. Overall, our results provide some interesting lessons about the field.

Two environmental communication journals, *Environmental Communication* and *Public Understanding*, are the primary publication destinations of climate-change-focused papers. Both journals are international, peer-reviewed publications indexed in Communication and Mass Media Complete and Science Citation Index Expanded (SCIE), two important indexes in communication and media studies and environmental science. They are both dedicated to climate change and sustainability, as well as media portrayals, public engagement and participation, and/or professional decisions. Articles often seek to bridge gaps between theory and practice, and are written in a style that is broadly accessible and engaging. Through our descriptive study, it was learned that, although topic-focused journals such as *Climatic Change* and *Journalism Studies* were ranked among the top five publication destinations of media-related climate change-focused articles, interdisciplinary publications were ranked highest. This is a positive result for media-related climate change because it is historically known that publishing in such journals helps students/academics to disseminate their articles to a much larger, scattered, and heterogeneous audience. There are

also more opportunities to adopt or combine different concepts, methods, and knowledge from two or more research areas.

The second lesson to be learned from this meta-analysis is that the media-centered climate change literature peaked in 2020 and has consistently grown over the last decade (2010-2020). This is consistent with, first, Schafer and Schlichting [14]'s study, who opined that "a look at the quantitative development of the research field indicates a clear growth: there has been a strong rise in scholarly attention for media coverage of climate change over the last few decades". Second, and probably the most significant reason for the upward trend of media-related climate change papers, which continues to attract scholarly attention, is that the media coverage and framing of climate change within global economies continues to be the gateway to climate change discussions. The number of articles remained moderate until 2013. From 2014 onwards, media-focused climate change publications increased significantly.

Lessons 3 and 5 from our media-centered climate change meta-analysis reveal that the United States and United Kingdom were the most studied countries and were also the countries with the most first or corresponding authors among the reviewed media-centered climate change articles. Apart from these countries, other developed European countries, such as Germany, were also significantly studied within the media-centered climate change literature. The pattern here shows that the opportunities afforded to researchers in the west are vast. Apart from access to funding from private and public organizations, and access to research institutes and personnel, there is ample access to information for media research, unlike in low-income countries suffering higher climate change impacts than the west. For example, India and Sri Lanka are ranked 18.17 and 19, respectively, on the Climate Risk Index, but coverage and access to information in such countries is limited when compared to the west. This is troubling. Keller et al. [17] state that the media-related climate change literature has "ignored other, equally important countries—namely developing countries and 'emerging economies' such as India. Understanding Indian media coverage will help understand evolving domestic agendas".

Although the aforementioned issue is a major challenge, our results also show that the media-related climate change literature has expanded in terms of geographical focus: there has been an increase in the number of countries studied. Spain, Uganda, Ireland, Poland, Pakistan, Croatia, Netherlands, Singapore, Tanzania, Switzerland, Belgium, Czech Republic, Turkey, Ghana, Denmark, Greenland, Chile, Vietnam, Malaysia, South Korea, Zimbabwe, Chile, and Peru were not studied based on the information available in Schafer and Schlichting [14]'s study.

Lesson 4 shows that media-centered climate change articles examined newspapers more than any other media form. The lesson here reinforces optimistic narratives about the future of newspapers. Schafer and Schlichting [14]'s study is consistent with our findings. Following their meta-analysis, it was found that similar media platforms were included among the dominant media adopted in the media-focused climate change literature, which seems striking. They stated that,

> *Probably the most striking finding of our meta-analysis is that more than two-thirds of all analyzed media (67.5%) are print media even though their proportion decreases over time. The share of print media was extraordinarily high in the early decades until the 1990s, when they accounted for more than 80% of all analyzed media, but shrank in the 2000s. Even then, however, print media still accounted for two-thirds of all analyzed media.*

While this is plausible for the traditional media optimists, it is also problematic in the sense that a large proportion of society today uses online platforms such as blogs, Facebook, Twitter, and Pinterest for information dissemination and consumption. In Ross and Rivers' study [7], they focused on climate change discourse within online memes, an informal yet powerful framing source for "political commentary, satire, and debates over notions of legitimacy". Their study did not directly provide new insights into researching memes as a communication tool, but introduced the possibility for researchers to look at climate issues from individual lenses, which sometimes void of an agenda and unconstrained

by ownership structure, the political economy of the Media Company, and advertising pressure, as with the traditional media companies. As Ross and Rivers [7] stated:

> *The ease with which memes are created and shared in relation to specific frames carries an important implication in that participation in debate and discussion in relation to social and political issues such as climate change becomes much more feasible and simple. For many, this participation might not extend beyond the creation (or even viewing) of a meme, but for many it might initiate an engagement that extends to more discursive interaction in new media, or even beyond that to other forms of activism, protest, or even to influencing voting choices that otherwise might not have emerged* (p. 991–992).

The review results showed higher percentages for newspapers as compared to six years ago, when the forenamed meta-analysis was conducted. Despite the overwhelming access to and popularity of new media technologies, the decline of newspapers or print media in general seems imminent. However, this study has shown and reinforced Schafer and Schlichting's [14] study that newspapers remain an important part of the media landscape. Schmidt and his colleagues [18] share the same idea that climate change coverage in newspapers has increased in all countries. According to Boussalis et al. [19] newspapers' coverage of climate change in Russia has risen steadily ever since the issue was identified as an international problem.

The lack of theory adoption within studies, as highlighted by our review, is not unique to this study. As a matter of fact, this is a problem that many media and/or communication and media studies and meta-analyses have acknowledged for a very long time. For example, in Zheng, Liang, Huang, and Liu's [20] meta-analysis on communication technologies, they found that "the major constraint on conducting communication technology research is a lack of theoretical orientation. Overall, theory-based communication technology research in Asia was rare and should be a key element of future endeavor". Among studies that adopted theories, we found that Framing Theory is the dominant theory used. "Frame theory and associated scholarship has served the domain of media and communication studies effectively for some time" [20]. Frames are extremely important in the discourse of climate change. As mentioned by Stecula and Merkley [21],

> *Frames related to climate change can emphasize economic costs or benefits, heighten partisan or ideological conflict, emphasize or downplay scientific uncertainty, among other things. There are likely implications for the public's support for climate action and willingness to act on these attitudes in a variety of ways—from voting for environmentally-friendly candidates to engaging in personal action to reduce one's own carbon footprint or even engaging in political activism* (p. 2).

Lessons 7 and 8 reveal that quantitative methods and content analysis are the most used methods and data collection tools within the media-centered climate change literature. Our review findings are also consistent with Schafer and Schlichting [14]'s, who opined that "approximately half of the publications use quantitative methods (47.8%), whereas 44.8% adopt a qualitative approach. While only 7.1% of all publications combine both research strategies in the same study, the respective trend points upward". Qualitative research and rhetoric analysis also attracted a large amount of scholarly attention. In parallel, Tillinghast and McCann [22] found that magazines underwent a shift from their initial episodic or isolated theme-oriented story structure to a broader and more connected thematic form. This is a positive result, considering that the two main paradigms are well represented. As reported by Schafer and Schlichting [14], "such a balance between the different paradigms and approaches should be welcome as it helps to balance out the complementary strengths and weaknesses of different approaches" (p. 992).

Finally, we observed that, although the distribution is almost equal, communication and media scholars are more interested in media-centered climate change research than environmental studies researchers. This is understandable because, as stated by Cohen [23], "the press may not be successful much of the time in telling people what to think, but it is stunningly successful in telling its readers what to think about". As in the case of

climate change, there is no better tool to inform people that this issue is one of the most important issues facing the human race. In the case of Bhutan, the whole country is committed to achieving Gross National Happiness (GNH) through sustainable environmental conservation and socioeconomic development; however, the country is facing increasing environmental challenges. Mongar [24] showed that the participants in his study were aware of various environmental problems; however, they lacked knowledge and awareness about climate change issues in Bhutan. Biswas [25] also drew attention to similar problems for the Bangladeshi case. Breslyn and McGinnis [26] considered computational thinking to provide climate change awareness in their study.

## 5. Conclusions

In conclusion, this systematic review has revealed important insights into the current state-of-the-art of the media-related climate change literature. We updated Schafer and Schlichting's [14] findings on the progression of the field, the analytical approaches, countries, types of media, and methodological approaches adopted within the field. This review also made new inquiries into other important questions, such as the theories adopted within the field, the first author's or corresponding author's department, the first author's or corresponding author's university's host country, publication venues, and the dominant subdiscipline of publication venues. Based on the results, it is possible to suggest that media-related climate change research activity has risen strongly over the years.

**Author Contributions:** Conceptualization, C.N.D.D. and A.G.; methodology, A.G.; software, C.N.D.D.; validation, C.N.D.D. and A.G; formal analysis, A.G.; investigation, C.N.D.D.; writing—original draft preparation, C.N.D.D..; writing—review and editing, A.G.; visualization, C.N.D.D.; supervision, A.G; All authors have read and agreed to the published version of the manuscript.

**Funding:** This research received no external funding.

**Institutional Review Board Statement:** Not applicable.

**Informed Consent Statement:** Not applicable.

**Data Availability Statement:** All data are included in the paper or could be openly achieved.

**Conflicts of Interest:** The authors declare no conflict of interest.

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
