# Peer review of "Lessons from Media-Centered Climate Change Literature"

_sustainability, doi:10.3390/su14031404_

Round 1

Reviewer 1 Report

This is an interesting study, fully suited to Sustainability journal.I have some ideas which can make the story more attractive for readers.

First authors university’s host country need not to be the only criterion which could be used – I am also sometimes moving my name at the end, but I am corresponding author, because I designed and wrote the MS. I think the authors need to check which percent of first authors were also corresponding authors. If in majority of cases corresponding authors were first and/or last corresponding author came in majority of cases from the same country as first author, then we can say that the analysis is unbiased. Please add these informaiton to the MS. 
L. 126 & 182: comfort and Park should be Comfort and Park
Table 1: Desenvolxvimento e meio ambiente should be Desenvolvimento e Meio Ambiente
I would remove a slope ʺValidʺ from tables, it looks redundant
Please make all letters in tables uniform in terms of their colour
It looks that the frequency of published papers about climate change continuously increases  - perhaps simple correlation between time and frequenty of these papers would be good to show to support this claim. 
For example, India and 372 Sri Lanka are ranked 18.17 and 19, respectively on the Climate Risk Index – it would be excellent to perform such analysis between climate risk index and number of papers about climate change (perhaps controlled with total number of people per country). 

Author Response

First of all thank you for your valuable comments. 

1- First authors university’s host country need not to be the only criterion which could be used – I am also sometimes moving my name at the end, but I am corresponding author, because I designed and wrote the MS. I think the authors need to check which percent of first authors were also corresponding authors. If in majority of cases corresponding authors were first and/or last corresponding author came in majority of cases from the same country as first author, then we can say that the analysis is unbiased. Please add these information to the MS. 

Reply: Yes i checked whether there is a difference in the analysis and looked how many of the first authors are at the same time correspondent authors. We realized that 150 out of 167 authors are at the same time either first or correspodent author this means that 89 percent of the analysis is same but since the related tables would change, I renewed the two related tables udated their numbers according to 150. And added the new info to MS in an unbiased way as you have noticed. Can see table 5 and table 9 in the revised fine, thanks.

2-L. 126 & 182: comfort and Park should be Comfort and Park
Table 1: Desenvolxvimento e meio ambiente should be Desenvolvimento e Meio Ambiente  I would remove a slope ʺValidʺ from tables, it looks redundant

Reply: All done, thanks, also extra proofreading done, can open track changes and have look as well.

3-Please make all letters in tables uniform in terms of their colour
Reply: Shading is also removed and all legends are in black for uniformity.

4-It looks that the frequency of published papers about climate change continuously increases  - perhaps simple correlation between time and frequenty of these papers would be good to show to support this claim. 
For example, India and 372 Sri Lanka are ranked 18.17 and 19, respectively on the Climate Risk Index – it would be excellent to perform such analysis between climate risk index and number of papers about climate change (perhaps controlled with total number of people per country). 

Reply: This will be done in future further research since it needs to be investigated further in detail and many variables would be needed, no time to check this statistically sorry.

Reviewer 2 Report

It is recommended to add, as an attachment, the list of papers used for the review.    

Author Response

Thank you for your recommendation, I have added the list of the papers as an attachment to the end pages as appendix. 

Reviewer 3 Report

General comment

This paper is in essence some literature review of the so-called “media related climate change centered literature.” Therefore, it is a “Review” paper instead of an “Article” which reports the results from scientific research.

Editorial comments

The manuscript needs a thorough editing because it contains typos, grammar errors and incomplete sentence. What follows are some of the examples:

(Lines 32-33) “… framing of the issue issues, people and events in and around climate change.” -> “issue issues” is an obvious typo.

(Line 33) “Corresponding to Wozniak, Lück & Wessler (2015),” -> This sentence is incomplete.

(Line 35) “The mass media have a mission …” “have” should be “has”.

(Line 50) “scholars interests both within media studies …” “scholars interests” should be “scholars’ ”.

Instead of providing a strong motivation statement, a very lengthy paragraph (lines 60-76) was used to state the nine subsections followed by a point summary. This just tells the readers how this review is organized, not an elaboration of what motivates this review.

(Line 94) “metanalysis” is a typo.

Method

The author(s) used frequency tables to summarize the results. The report of frequency tables is a way of presenting descriptive summary, not “empirical evidence” as mentioned in the paper.

Author Response

First of all thank you for your contributions.

The manuscript needs a thorough editing because it contains typos, grammar errors and incomplete sentence. What follows are some of the examples:

(Lines 32-33) “… framing of the issue issues, people and events in and around climate change.” -> “issue issues” is an obvious typo.  =>Corrected

(Line 33) “Corresponding to Wozniak, Lück & Wessler (2015),” -> This sentence is incomplete. Corrected

(Line 35) “The mass media have a mission …” “have” should be “has”. => Corrected

(Line 50) “scholars interests both within media studies ”“scholars interests” should be “scholars’ ” =Corrected

Instead of providing a strong motivation statement, a very lengthy paragraph (lines 60-76) was used to state the nine subsections followed by a point summary. This just tells the readers how this review is organized, not an elaboration of what motivates this review.    Corrected and summarized better with removing the numbers in the paragraph.

(Line 94) “metanalysis” is a typo. => Corrected

Method

The author(s) used frequency tables to summarize the results. The report of frequency tables is a way of presenting descriptive summary, not “empirical evidence” as mentioned in the paper.

  • Done Line, 77 , 297, 307, 316 and 344 empirical evidence is corrected descriptive summary added/Modified .

You are right, it is a systematic review and done in a descriptive methodology. Correcting this is done via mentioning it as study in abstract and turned into a systematic review paper. As it is corrected in methodology, also requested to appear like review paper. Wording is changed as well like either review or study. Extra proofreading done see track changes please.

Also third person language is used instead of `We` sentences

Round 2

Reviewer 3 Report

As suggested in my first-round review which the authors agreed on, this paper should be submitted as a “Review” instead of an “Article.”

I did not find significant improvement in the writing of the manuscript. The revised manuscript needs still needs an extensive editing because it contains typos, grammar errors and incomplete sentences.

I have some further comments on the revised version. Note that what follows are just some of the typos and sentences that are incomplete or contain grammar errors.

(Line 25) “the discourse of climate is inconceivably in the first instance,” -> “inconceivably” is an adverb while an adjective should be used here.

(Lines 28-29) “Many of above and beyond affect the everyday language of the everyday people.” What is this sentence meant to express?

(Line 44) “As acknowledged many scholars” -> “by” is missing here.

(Lines 49-50) “Internet memes and climate change (Ross and Rivers, 2019), has received scholars’ both within media studies …” -> Some word is missing, say “attention”, following scholars’?

(Lines 50-51) “Although media and climate change cannot be considered as a distinct area of disci-pline of its own yet.” -> “of its own” should be “of their own”. Moreover, this sentence is incomplete.

(Line 56) “metanalysis of climate change” -> As mentioned in my first-round review, “metanalysis” is a typo.

(Lines 57-59) “And finally, be-cause of the ongoing coverage of climate change across various media platforms across the world and continuous publication of the media related issues within climate change.” -> This sentence is incomplete.

(Line 71) “correspondent authors` university’s host country” -> Should be rephrased as “the host country of correspondent author's university“.

(Lines 112-113) “And, although there isn’t a generally accepted way of conducting of systematic literature review, we adopt Denyer and Tranfield, (2009) method a systematic literature …” -> There is a grammar error in “a generally accepted way of conducting of”. Moreover, some word is missing in “we adopt Denyer and Tranfield, (2009) method a systematic literature”.

Author Response

Dear Reviewer,

first of all thank you for your constructive comments, extensive editing is done and a good proofreading is carried out with the help of the MDPI English Services. Also for the scientific part we added 5 new references in order to better cover and discuss the literature.
